# Cocoa Bar Antioxidant Profile Enrichment with Underutilized Apples Varieties

**DOI:** 10.3390/antiox11040694

**Published:** 2022-03-31

**Authors:** Alessandra Francini, Carmen Fidalgo-Illesca, Andrea Raffaelli, Marco Romi, Claudio Cantini, Luca Sebastiani

**Affiliations:** 1BioLabs, Institute of Life Sciences, Scuola Superiore Sant’Anna, Piazza Martiri della Libertà 33, 56127 Pisa, Italy; alessandra.francini@santannapisa.it (A.F.); c.fidalgoillesca@santannapisa.it (C.F.-I.); andrea1.raffaelli@santannapisa.it (A.R.); 2Department of Life Sciences, University of Siena, Via Mattioli 4, 53100 Siena, Italy; marco.romi@unisi.it; 3Institute for BioEconomy, National Research Council of Italy, 58022 Follonica, Italy; claudio.cantini@ibe.cnr.it

**Keywords:** antioxidant, polyphenols, enriched foods, DPPH, old variety

## Abstract

The impact of dried apples (*Malus* × *domestica* Borkh.) addition on improving the antioxidant characteristics of dark chocolate was evaluated. The antioxidant activity was measured through DPPH scavenging activity and showed an increase in the cocoa bar with ‘Nesta’ dry apple (17.3% vs. 46.8%) in comparison to cocoa mass. The 15 polyphenols analyzed by UHPLC-ESI-MS/MS indicated great variability among the apple varieties. Quercetin was detected in the highest concentrations (ranged from 753.3 to 1915.5 µg g^−1^), while the lowest were for kaempferol 7-*O*-glucoside, measured only in ‘Mora’ and ‘Nesta’ cocoa bars (from 0.034 to 0.069 µg g^−1^, respectively). *P*-coumaric acid, *trans*-ferulic acid, and chlorogenic acid contribute largely to the antioxidant activity in cocoa bars. Principal component analysis shows that a cocoa bar with the addition of ‘Nesta’ dry apple differ from others due to its higher content of polyphenols (1614 ± 61.8 mg gallic acid equivalents per 100 g). In conclusion, data confirm that cocoa bars with dry apples might be considered as a polyphenol-enriched food.

## 1. Introduction

Food enrichment is a widely used process aimed to modify the composition and content of nutrients and antioxidants in food products. The purpose is to increase their nutritional and nutraceutical quality, meet consumers’ expectation, and increase market demand.

Many fruits are rich in nutrients and nutraceutical compounds; e.g., apple (*Malus* × *domestica* Borkh.) fruits and derived products are particularly rich in antioxidant molecules [1]. Dried apples have a high concentration of minerals, vitamins, and polyphenols [1,2]. For this reason, apples can be used for the preparation of enriched foods in many forms: fresh, dry, as juice extract, and even as apple pomace by-product [3,4,5]. It is noteworthy that desiccation of apples rich in polyphenols may cause a dramatic change in color, making them brown and not attractive for the consumers. In turn, a dark appearance during the desiccation process could be a positive trait when dried apple slices or chips are mixed with cocoa.

Many underutilized apple varieties show a fibrous peel, irregular size, and astringency that make them below the commercial acceptability limits for fresh markets. On the contrary, they have a higher concentration of biologically active compounds when compared to commercial cultivars [6,7,8], making underutilized apple germplasm interesting from a nutraceutical point of view [9,10]. Their combination with cocoa might enhance the marketing of defective-in-shape or colored apple fruits not good enough for fresh use but still interesting for their nutritional and nutraceutical potential.

Cocoa is considered one of the most important agricultural products worldwide. It is a product with high nutritional value and beneficial for health. Moreover, the consumer appreciates cocoa sensorial characteristics. Polyphenols present in cocoa powder are considered relevant in this product [11,12]; among them, flavanols have beneficial effects on health and aging, improving blood pressure, flow-mediated dilation, and vascular stiffness in healthy subjects [13].

The consumption of high-quality cocoa products and chocolates enriched with dry fruits are increasing. It has been shown that dry apples affect the chocolate’s texture and have an impact on consumer acceptability of the cocoa bar [3]. Less known in the literature is the effective improvement of cocoa nutraceutical profile with polyphenols derived by the addition of dry apple fruits. Apples polyphenols are interesting considering their ability to modulate several cell-signaling pathways in a wide range of human pathologies [14,15,16,17]. As an example, apples have a high concentration of quercetin, which is a powerful antioxidant that promotes brain growth, stimulates dopamine levels, and is considered very important in preventing diseases such as lung cancer and type II diabetes [18].

The aim of this work is to study the polyphenol profile of a new set of cocoa bars—named Toscolata^®^—containing dried apples prepared from underutilized and commercial varieties. Polyphenol identification and concentration changes were studied using a liquid chromatography–electrospray ionization mass spectrometry (UHPLC-ESI-MS/MS) approach.

## 2. Materials and Methods

### 2.1. Chemicals

*P*-coumaric acid (PCA), *trans*-ferulic acid (TFRA) phloretin (PHL), (+)-catechin (CTC), (-)-epicatechin (ECTC), quercetin (QCT), chlorogenic acid (CGA) phloridzin (PDZ), kaempferol 7-*O*-glucoside (KPF7G), kaempferol 3-*O*-glucoside (KPF3G), quercetin 3-*O*-glucoside (QCT3G), quercetagetin 7-*O*-glucoside (QGT7G), tiliroside (TSD), Kaempferol 3-*O*-rutinoside (KPF3R), rutin (RTN), HPLC grade methanol, Folin–Ciocalteu’s phenol reagent, gallic acid, acetonitrile, Na_2_CO_3_, formic acid, and 1,1-Diphenyl-2-picrylhydrazyl were purchased from Sigma–Aldrich (Milan, Italy). The aqueous solutions were prepared with ultra-pure water purified by a Milli-Q System (Millipore, Milan, Italy).

### 2.2. Materials

Five underutilized apples cultivars, namely, ‘Mora’, ‘Nesta’, ‘Ruggina’, and ‘Panaia’, were used. We used as the reference standard the cultivar ‘Golden Delicious’. The latter was selected because it is widely cultivated and studied in the literature. Apple fruits were harvested at commercial ripening stage. This stage was determined on the base of several fruit parameters as reported in Francini et al. [2]. After washing, the core of each apple was removed, and the flesh cut into thick pieces. The dehydration was performed by an air-drying oven set at a steady temperature of 35 °C for 10 h. At the end of the drying process, the slices were added to the cocoa bar prototypes (2.5%), as described in Cantini et al. [3].

### 2.3. Polyphenols Extraction

Aqueous methanol (20:80, *v*:*v*) was used for polyphenols extraction using 25 mL for 7.5 g of the following samples (*n* = 3): (a) dry apples; (b) dark cocoa bar (70%); (c) dark cocoa bar (70%) containing 2.5% of dry apples. Total phenols content was determined in technical triplicates following the reduction of Folin–Ciocalteau reagent by phenolic compounds, with a formation of a blue-colored complex. Samples (30 μL) were added to Folin–Ciocalteu’s phenol reagent (150 μL), or standard solution of gallic acid, and the mixture was shaken. After 10 min, 600 μL of a 2% (*w*/*v*) Na_2_CO_3_ solution was added then diluted to 3 mL with Milli-Q ultra-pure water, mixed, and finally incubated for 30 min at 70 °C. The absorbance of the solution was measured at 750 nm using water as blank control. A standard curve of gallic acid solutions in the range of 0.070–2.50 mg mL^−1^ was prepared to determine the total phenolic content of the samples, expressed as mg gallic acid equivalents per g of material.

### 2.4. DPPH Assay

Following the protocol of Francini et al. [19], the scavenging activity of 1,1-diphenyl-2-picrylhydrazyl (DPPH) radicals was evaluated. Briefly, 200 μL of the methanol extract of the cocoa bar and the cocoa bar with dry apples were mixed with 800 μL of a Tris-HCl 100 mM solution, pH 7.0. Samples were then supplemented with 250 µM DPPH and kept in the dark for 30 min. Methanol was used as control reference and absorbance measured at 517 nm. DPPH inhibition (%) was calculated as follows: Inhibition ratio (%) = ((absorbance of control − absorbance of sample) (absorbance of control)) × 100.

### 2.5. UHPLC-ESI-MS/MS Analysis

Selected known polyphenols were chosen in this study and quantitative analyses were done in the extracts by UHPLC-ESI-MS/MS using a Sciex 5500 QTrap+ mass spectrometer (AB Sciex LLC, Framingham, MA, USA), equipped with a Turbo V ion-spray source and coupled to an ExionLC AC System custom made by Shimadzu (Shimadzu Corporation, Kyoto, Japan) which includes 2 ExionLC AC Pumps, Autosampler, Controller, Degasser and Tray.

MS/MS experiments were performed in the electrospray negative ion mode using nitrogen as collision gas. The operation source parameters were source type, Turbospray; nebulizer gas (GS1) 70; turbo gas (GS2) 50; curtain gas (CUR) 10; temperature (TEM) 500 °C; Ionspray Voltage (IS) −4500 V, entrance potential (EP) 10 V. Compound parameters, declustering potential (DP), collision energy (CE), and collision cell exit potential (CXP) were adjusted for the specific Selected Reaction Monitoring (SRM) transition for any component. SRM transitions and the corresponding compound parameters are reported in Table 1.

Chromatographic separation was performed as reported in Francini et al. [20] by a Phenomenex Kinetex EVO 2 × 100 × 5 µm column (Phenomenex, Torrance, CA, USA). The elution was carried out in gradient mode using acetonitrile containing 0.1% formic acid (solvent A) and water containing 0.1% formic acid (solvent B). A gradient elution was programmed: 0.0 min, A 5%; 0.0–10.0 min, A 5–95%; 10.0–12.0 min, A 95%; followed by 4 min equilibration time (A 5%). The flow rate was 300 µL min^−1^, injection volume 20 µL, and column oven temperature 40 °C.

The recovery percentage and matrix effect were used to normalize the data. Recovery was calculated as the peak area of the sample spiked before extraction/peak area of the sample spiked after extraction, while the matrix effect was calculated as the peak area of the sample spiked after extraction/peak area of the standard.

### 2.6. Statistical Analyses

Data are shown as the mean ± standard deviation of three-replicates. One-way analysis of variance (one-way ANOVA) was performed in Graph Pad Prism 6.0 (GraphPad software, San Diego, CA, USA) and Tukey’s post-hoc test was further applied when *p* ≤ 0.05 using the same statistical package. Principal component analysis was performed with a statistical software program (NCSS 2004, NCSS Statistical Software, Kaysville, UT, USA).

## 3. Results

The dry apples used for the cocoa bar production, the cocoa bar with dry apple, and the cocoa mass were analyzed for DPPH radical scavenging capacity and total polyphenols (Table 2).

In dry apples, a lower DPPH activity was detected in the variety ‘Mora’ (22%). The other apple showed a stronger activity than Mora, with the higher values for ‘Nesta’ (47.6%) and ‘Panaia’ (66.6%) (Table 2). In a cocoa bar with dry apple, the DPPH activity decreased in many combinations, except in ‘Mora ‘and ‘Nesta’.

Regarding the total phenolic content of the dry apples, ‘Nesta’ is the variety with the higher total phenolic content (1614 mg GAE/100 g of dry weight) while ‘Golden’ and ‘Ruggina’ had the lowest values (991 and 927 mg gallic acid equivalents per 100 g of DW respectively). A significant increase in the DPPH radical scavenging capacity was shown when comparing the cocoa mass and cocoa bar with ‘Nesta’ dry apple (17.3% vs. 46.8%).

Data of total phenolic content reached the highest value in the cocoa bar with ‘Nesta’: +71% when compared to the cocoa mass (Table 2).

On the basis of previous investigation, a panel of 15 polyphenols were selected (Figure 1): 7 flavonols (quercetagetin 7-*O*-glucoside, rutin, quercetin 3-*O*-glucoside, kaempferol 3-*O*-rutinoside, kaempferol 7-*O*-glucoside, kaempferol 3-*O*-glucoside and quercetin), 2 flavanols (epicatechin and catechin), 1 cinnamate ester (chlorogenic acid), 2 hydroxy-cinnamic acids (*p*-coumaric and *trans*-ferulic acid), 2 dihydrochalcones (phloretine and phloridzin), and 1 oxyflavone (tiliroside).

A single SRM transition was used for the UHPLC-ESI-MS/MS analysis and for molecule quantification. Qualitative confirmation was acquired taking advantage of one of the features of a QTrap instrument. Information Dependent Acquisition (IDA) was programmed so that the SRM transitions reported in the Table 1 were used as a survey scan, switching the third quadrupole to act as a Linear Ion Trap, performing an Enhanced Product Ions scan, affording the complete MS-MS product ions spectrum (MRM >> Enhanced Product experiment). A comparison with a custom-built MS-MS product ions spectra library allowed the qualitative confirmation.

All the phenolic compounds showed great variability among varieties (Figure 1). The higher content was detected for quercetin, ranging from 2256 to 255 µg g^−1^ DW (Figure 1A), while the lower for kaempferol 3-*O*-rutinoside, ranging from 0.006 to 0.143 µg g^−1^ DW in ‘Panaia’ and ‘Ruggina’, respectively (Figure 1O). The flavanol epicatechin and the flavonol quercetagetin 7-*O*-glucoside did not showed difference among the dry apples studied (Figure 1B,K). The higher significant value of PCA and TFRA was observed in ‘Ruggina’ extract (1.02 and 0.49 µg g^−1^ DW, respectively), while the higher significant value of RTN and KPF3R in ‘Panaia’ dry apples (14.7 and 0.14 µg g^−1^ DW, respectively). The reference ‘Golden’ apple showed higher value of QCT and KPF3G (Figure 1A,I).

In Figure 2, the specific phenolic composition of the cocoa bar with dry apples are presented together with the cocoa mass.

Cocoa bar with ‘Nesta’ has the highest concentrations of almost all the 15 polyphenols under study. Moreover, all cocoa bars with dry apples showed a significant increase in the PDZ and CGA concentrations when compared to cocoa mass. Quercetin was the more abundant polyphenol (Figure 2A), while kaempferol 7-*O*-glucoside was less abundant, being detected only in cocoa bars with ‘Mora’ and ‘Nesta’ and ranged from 0.034 to 0.069 µg g^−1^, respectively (Figure 2O).

A principal component analysis was carried out with the aim of highlighting the relationship among selected variables and cocoa bars with dried apples and cocoa mass (Figure 3). Using the 15 polyphenols, the two principal components explain 84.11% of total variance, and allow a good differentiation among cocoa bars and cocoa mass.

The variables that contributed more to PC1 (57.6%) were the *trans*-ferulic acid, phloretin, (+)-catechin, (−)-epicatechin, quercetin, chlorogenic acid, phloridzin, kaempferol 3-*O*-glucoside, quercetin 3-*O*-glucoside, kaempferol 3-*O*-rutinoside and rutin. The PC2 (26.5%) was associated with *p*-coumaric acid, kaempferol 7-*O*-glucoside, quercetagetin 7-*O*-glucoside, and tiliroside (Table 3). Phloretin and quercetin 3-*O*-glucoside contributed most to explain the variation in the first component, followed by quercetin, while in the second component *p*-coumaric acid and quercetagetin 7-*O*-glucoside were the most discriminant variables.

Data indicated that there was a significant differentiation in three groups among the cocoa bars (Figure 3). According to the PCA biplot, the cocoa bar with ‘Nesta’ dry apple had the higher polyphenol concentrations; the cocoa bar with ‘Golden’ dry apples showed an intermediate behavior, while the other cocoa bars containing ‘Mora’, ‘Panaia’, and ‘Ruggina’ are similar to the cocoa mass.

Positive correlations between the phenolic compound concentration and total antioxidant capacity (DPPH) in the cocoa bars with dry apples were found (Table 4). All polyphenols positively contributed to the antioxidant activity in the cocoa bar, except KPF3G and TSD.

## 4. Discussion

Following the increased demand for foods with nutraceutical value added, this research explores the possibility to add antioxidant compounds from dry apples to cocoa mass. Flòrez-Mendéz et al. [21] proved the importance of the consumption of chocolate enriched with tryptophan and resveratrol, showing a significant decrease in insulin in blood after consuming 15 g of chocolate for 21 days, making it a functional food. Our previous studies also have noted the beneficial effects on human health of cocoa enriched with other fruit products [22].

Many companies are now considering the incorporation into chocolate formulations of functional ingredients, to gratify consumer demands from both a health and sensory point of view. The development of new chocolate formulations with a healthier profile [23] indicate that the added ingredients could affect the sensory properties of the original products. To this respect, panel tests and consumer acceptability of cocoa bars improved with Tuscan autochthonous food products [3], demonstrating that cocoa bar with apples have a sensory profile widely appreciated by both professional panelist and consumers.

The reason for studying the bioactive polyphenolic compounds in underutilized varieties derived from the observation that, in several cases, they are rich of polyphenols [1]. Using state-of-the-art UHPLC-ESI-MS/MS analytical tools, we proved that apple varieties have a great variability among the fifteen phenolic compounds analyzed. This variation among polyphenols is explained in the literature in relation to resistance to diseases or adverse environmental conditions between varieties [24]. Moreover, underutilized apple varieties are often characterized by astringency, which correlate with high polyphenols concentrations [25], and is a trait that in commercial varieties has been slightly reduced, favoring others such as sweetness and lower acidity.

In apples, flavonoids, including flavones, flavanols, anthocyanins, and dihydrochalcones, are abundant [26], and quercetin, (−)-epicatechin, (+)-catechin, and chlorogenic acid are besides these the main polyphenols [1].

Researchers have investigated the anti-inflammatory activity of phloridzin and phloretin [27,28,29], demonstrating that these compounds present in apples could be positively linked with the beneficial effects of apple consumption. It is interesting to note that in cocoa bars with dry apples a significant increment of phloridzin has been demonstrated and also that cocoa bars with ‘Nesta’ and ‘Mora’ have a three-times higher value of phloretin compared to cocoa mass.

During cocoa bar preparation, the concentration of total and single bioactive compounds could be significantly modified [30]. The polyphenols concentration measured in cocoa bars with dry apples products is not simply a sum between the molecules in dry apples and cocoa mass. The final concentration is strictly related to the transformations occurring to single polyphenols during the manufacturing process in each cocoa bar with dry apple combinations. The literature reports that, even in chocolate alone, an industrialization step reduces the total polyphenols originally present in cocoa bean. For example, polyphenol oxidization reduce the flavanol content and determine the degradation of specific polyphenols such as (−)-epicatechin and (+)-catechin. Regarding flavonol, quercetin undergoes the greatest decrease during the industrialization processes [31]. Gültekin-Ozgüven et al. [32] demonstrated that, during chocolate manufacturing, a reduction in phenolic content is normally expected due to high temperatures. Moreover, chemical transformation like epimerization of the (−)-epicatechin in (−)-catechin could occur [33]. The complexity of these transformations during thermal processing to which polyphenols undergo may significantly differ when polyphenols derive not only from cocoa but also from apple. Additional aspects are related to the presence of vitamins in dry apples that could have a role in preserving polyphenol reduction during cocoa bar preparation.

In general, these data demonstrate the importance of using dry apple material with the highest concentration of polyphenols, thus minimizing total and specific antioxidant reductions when dry apples are added to the cocoa bars. These complex interactions during cocoa bar with dry apple preparation could determine the quick reduction in specific polyphenols such as quercetin, with differences between ‘Golden’ and the other apple varieties.

Concerning the DPPH scavenging activity, statistical analyses of our data indicated that all the dry apples (except ‘Mora’) have a statistically similar DPPH activity. After cocoa bar preparation, this activity was reduced in ‘Golden’ (−26.4%), ‘Panaia’ (−63.7%), and ‘Ruggina’ (−45.6%), while remaining stable in ‘Mora’ and ‘Nesta’ samples. The different reductions in DPPH activity, despite showing no significant differences in DPPH at the dry apple levels, might be due to the presence of vitamins and the polyphenol profile of the apples studied. For all these reasons, our data prove that it is very important to provide a real measure of the polyphenols and DPPH scavenging activity in the final products.

The relationship between bioactive compounds and the sensory properties of dark chocolate produced from Brazilian hybrid cocoa have been demonstrated by das Virgens et al. [34]. The higher contents of epicatechin, catechin, and total phenolic compounds contributed to a higher intensity of bitterness, cocoa flavor, acid taste, and astringency. The research demonstrates the importance of the improvement of these compounds for consumer satisfactoriness. Moreover, the significant positive correlations found between phenolic compound concentrations, such as *p*-coumaric acid, *trans*-ferulic acid, and chlorogenic acid, and total antioxidant capacity in cocoa bars indicated their role as molecule scavengers of the oxidant compounds. This was particularly evident for the ‘Nesta’ variety apples, which were well separated using principal component analysis in relation to the higher content of polyphenols detected in dry apples and transferred to the cocoa bars.

## 5. Conclusions

Underutilized apple varieties could be added to fine cocoa, modifying the chemical and sensory value of the final product. These changes could successfully increase the consumers’ interest in these new cocoa-based foodstuffs. The UHPLC-ESI-MS/MS approach developed in this study is a strategic tool for cocoa producers to achieve this objective and provide a real quantification of the polyphenols in the final products.

## Figures and Tables

**Figure 1 antioxidants-11-00694-f001:**
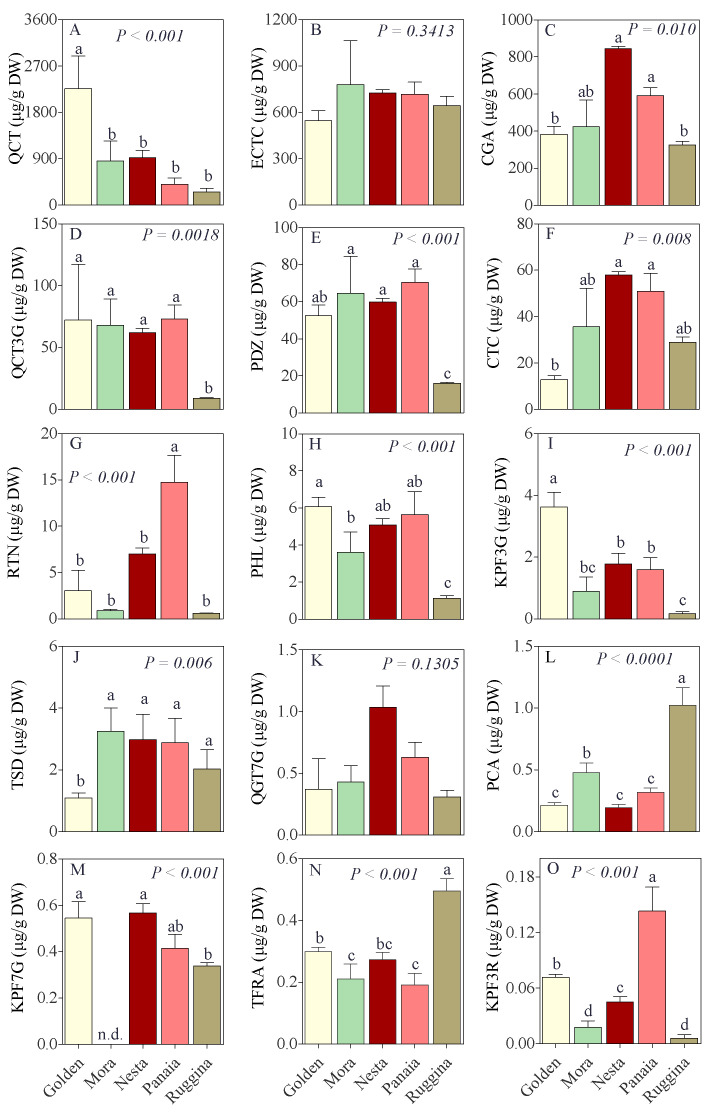
Mean values and standard deviation for the concentration (μg/g DW) of (**A**) quercetin (QCT), (**B**) (−)-epicatechin (ECTC), (**C**) chlorogenic acid (CGA), (**D**) quercetin 3-*O*-glucoside (QCT3G), (**E**) phloridzin (PDZ), (**F**) (+)-catechin (CTC), (**G**) rutin (RTN), (**H**) phloretin (PHL), (**I**) kaempferol 3-*O*-glucoside (KPF3G), (**J**) tiliroside (TSD), (**K**) quercetagetin 7-*O*-glucoside (QGT7G), (**L**) *p*-coumaric acid (PCA), (**M**) kaempferol 7-*O*-glucoside (KPF7G), (**N**) *trans*-ferulic acid (TFRA), and (**O**) kaempferol 3-*O*-rutinoside (KPF3R), determined by UHPLC-ESI-MS/MS for dry apples. Data (*n* = 3) were analyzed by one-way ANOVA. Different letters are significantly different according to Tukey’s post-hoc test at the 0.05 probability level.

**Figure 2 antioxidants-11-00694-f002:**
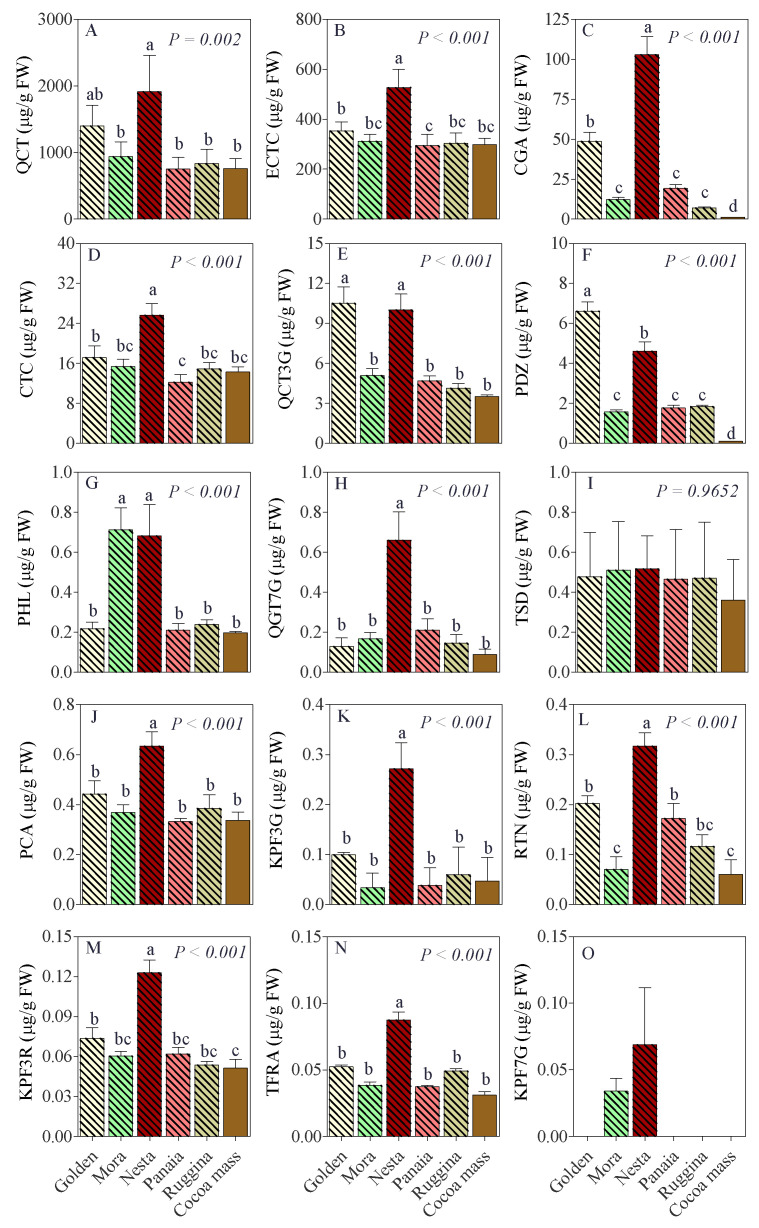
Mean values and standard deviation of (**A**) quercetin (QCT), (**B**) (−)-epicatechin (ECTC), (**C**) chlorogenic acid (CGA), (**D**) (+)-catechin (CTC), (**E**) quercetin 3-*O*-glucoside (QCT3G), (**F**) phloridzin (PDZ), (**G**) Phloretin (PHL), (**H**) quercetagetin 7-*O*-glucoside (QGT7G), (**I**) tiliroside (TSD), (**J**) *p*-coumaric acid (PCA), (**K**) kaempferol 3-*O*-glucoside (KPF3G), (**L**) rutin (RTN), (**M**) kaempferol 3-*O*-rutinoside (KPF3R), (**N**) *trans*-ferulic acid (TFRA), and (**O**) kaempferol 7-*O*-glucoside (KPF7G), expressed as μg/g FW for cocoa bars and cocoa mass determined by UHPLC-ESI-MS/MS. Data (*n* = 3) were analyzed by one-way ANOVA. Different letters are significantly different according to Tukey’s post-hoc test at the 0.05 probability level.

**Figure 3 antioxidants-11-00694-f003:**
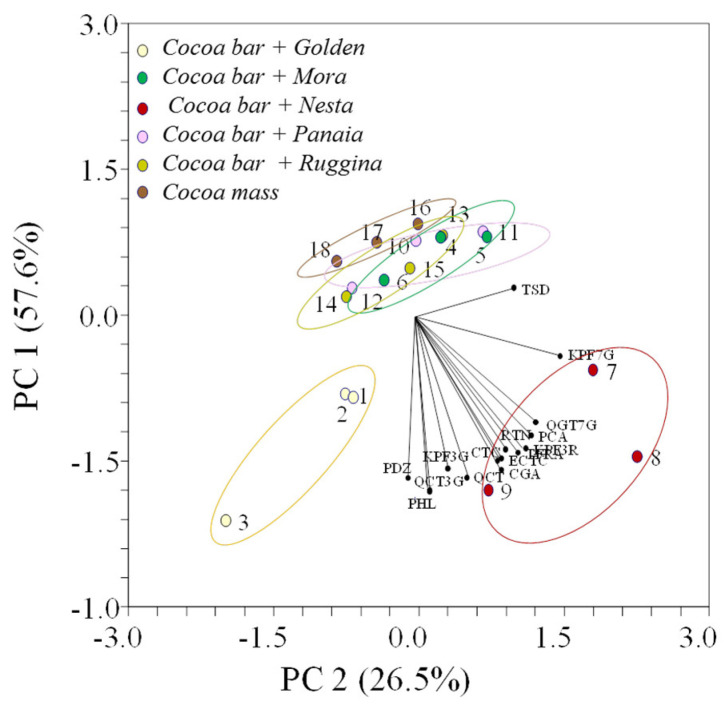
Biplot of PC1 vs. PC2, showing the loadings of the scores and the variables of cocoa mass and the cocoa bars with apples. Color dots with the number representing different cultivars analysed in triplicate.

**Table 1 antioxidants-11-00694-t001:** Relative polyphenol compound parameters, retention time (RT), and selected reaction monitoring transitions.

Name	Acronym	RT (min)	Q1	Q3	DP (V)	CE (eV)	CXP (V)
Catechin	CTC	2.33	289.0	244.9	−108	−22	−11
Chlorogenic acid	CGA	2.44	353.0	191.0	−61	−24	−9
Epicatechin	ECTC	2.63	289.0	244.9	−108	−22	−11
Quercetagetin 7-*O*-glucoside	QGT7G	3.07	479.1	316.9	−152	−31	−14
*p*-coumaric acid	PCA	3.09	163.0	119.0	−65	−18	−11
*trans*-ferulic acid	TFRA	3.28	193.0	134.0	−62	−20	−8
Rutin	RTN	3.30	609.2	299.9	−154	−48	−11
Quercetin 3-*O*-glucoside	QCT3G	3.40	463.1	300.0	−154	−37	−5
Kaempferol 3-*O*-rutinoside	KPF3R	3.53	593.2	284.9	−138	−40	−5
Kaempferol 3-*O*-glucoside	KPF3G	3.62	447.1	284.1	−202	−39	−11
Kaempferol 7-*O*-glucoside	KPF7G	3.66	447.1	284.9	−158	−38	−5
Phloridzin	PDZ	3.71	435.1	272.9	−135	−23	−5
Quercetin	QCT	4.30	301.0	150.9	−113	−38	−8
Tiliroside	TSD	4.41	593.2	284.9	−138	−40	−5
Phloretin	PHL	4.64	273.0	167.0	−103	−38	−11

**Table 2 antioxidants-11-00694-t002:** Total antioxidant capacity (DPPH%) and total polyphenols concentration (mg gallic acid equivalents per 100 g of DW or FW) in dry apples, in the cocoa bar with dry apples, and in the cocoa mass. Data (*n* = 3) were analyzed by one-way ANOVA. Different letters are significantly different according to Tukey’s post-hoc test at the 0.05 probability level.

		DPPH (%)	Total polyphenols (mg Gallic Acid Equivalents per 100 g)
Dry apple	Golden	50.7 ± 5.28 a	991 ± 80.4 c
	Mora	22.7 ± 3.39 b	1473 ± 18.6 b
	Nesta	47.6 ± 5.80 a	1614 ± 61.8 a
	Panaia	66.6 ± 8.56 a	1428 ± 57.1 b
	Ruggina	55.1 ± 2.93 a	927 ± 16.2 c
Cocoa bar + Dry apple	Golden	37.3 ± 14.54 ab	640 ± 151.4 b
	Mora	27.1 ± 1.14 ab	563 ± 46.8 b
	Nesta	46.8 ± 4.14 a	890 ± 85.9 a
	Panaia	24.2 ± 6.28 b	502 ± 34.0 b
	Ruggina	30.0 ± 4.04 ab	561 ± 33.2 b
Cocoa mass		17.3 ± 3.80 b	519 ± 23.9 b

**Table 3 antioxidants-11-00694-t003:** Loadings of each attribute on the two principal components calculated using the data produced by the UHPLC-MS/MS on six cocoa bars.

Variables	Factor 1	Factor 2
PCA	−0.6561	0.6776
TFRA	−0.7522	0.6011
PHL	−0.9715	0.0863
CTC	−0.7986	0.4807
ECTC	−0.7853	0.5045
QCT	−0.8925	0.3033
CGA	−0.8483	0.5050
PDZ	−0.8940	−0.0402
KPF7G	−0.2165	0.8474
KPF3G	−0.8424	0.1918
QCT3G	−0.9629	0.0860
QGT7G	−0.5897	0.7203
TSD	0.1636	0.5776
KPF3R	−0.7284	0.6470

**Table 4 antioxidants-11-00694-t004:** Pearson’s correlation coefficient between the polyphenol concentration and the total antioxidant capacity (DPPH) on cocoa bar with dry apples. Significance levels expressed by asterisks (*** for *p*-value ≤ 0.001; ** for *p*-value ≤ 0.01; * for *p*-value ≤ 0.05). n.s.: not significant.

Phenolic Compound	Correlation Coefficient
CTC	0.401 (**)
CGA	0.569 (***)
ECTC	0.356 (*)
QGT7G	0.347 (*)
PCA	0.650 (***)
TFRA	0.639 (***)
RTN	0.601 (***)
QCT3G	0.475 (**)
KPF3R	0.585 (***)
KPF3G	0.144 (n.s.)
KPF7G	0.525 (*)
PDZ	0.465 (**)
QCT	0.325 (*)
TSD	0.088 (n.s.)
PHL	0.452 (**)

## Data Availability

Data is contained within the article.

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
