# Peer review of "Cocoa Bar Antioxidant Profile Enrichment with Underutilized Apples Varieties"

_antioxidants, 2022, doi:10.3390/antiox11040694_

Round 1

Reviewer 1 Report

Comments:

This work investigated the impact of dried apples (Malus x domestica Borkh.) addition on the improvement of antioxidant characteristics of dark chocolate. DPPH free radical scavenging assay has been used to detect the antioxidant activity of samples.  UHPLC-ESI-MS/MS was used for the qualitative and quantitative analysis of polyphenols in the samples. They found dry apple addition to dark chocolate could improve the polyphenols concentration and antioxidant capacity of the cocoa bar.  Dry apple prepared from the cultivar ‘Nesta’ was significantly different from other cultivars. The cocoa bar with dry ‘Nesta’ apple might be considered as an enriched food due to both newly added compounds brought by apple and the increase of those naturally present on apple and cocoa itself. This is an interesting work. On one hand, it is a good way to enrich the nutrients of chocolate products with fruits. In another hand, the underutilized apple varieties that have been used can avoid the waste of natural resources. However, the organization of the manuscript must be improved before it can be published.

  1. The discussion section is too weak, I cannot find any answer to my question which was raised when I read the result part. For example, the content of quercetin in cultivar ‘Golden’ is much higher than other cultivars, why the cocoa bar added to this cultivar is not the highest one? In the same way, for DPPH scavenging activity, the ‘Panaia’ is the best one, but with cocoa mass, its activity became the weakest one? There are many questions between the compounds and activities. I think the authors need to think about those questions and try to give a reasonable explanation.
  2. line 73, the first letter should be capital. The abbreviation of p-coumaric acid (PCA) could cause confusion with Principal Component Analysis (PCA) used later, please try to make it clearer. In my eyes, all the abbreviations used here for the chemical compounds are very strange and difficult for remembering.
  3. Line 139-140, why the daughter ion of 299.9 was used for rutin? From my experience, the most abundant daughter ion for rutin would be 301. For the quercetin-3-O-glucoside, it is 301 too. For kaempferol 3-glucoside, it is around 285, like other glycosides of kaempferol.

Author Response

COMMENT #1. The discussion section is too weak, I cannot find any answer to my question which was raised when I read the result part. For example, the content of quercetin in cultivar ‘Golden’ is much higher than other cultivars, why the cocoa bar added to this cultivar is not the highest one? In the same way, for DPPH scavenging activity, the ‘Panaia’ is the best one, but with cocoa mass, its activity became the weakest one? There are many questions between the compounds and activities. I think the authors need to think about those questions and try to give a reasonable explanation.

Answer: Following the Comment of Reviewer 1 some improvement of the discussion section have been done and new references has been introduced (see manuscript changes in red).

COMMENT #2. Line 73, the first letter should be capital.

The abbreviation of p-coumaric acid (PCA) could cause confusion with Principal Component Analysis (PCA) used later, please try to make it clearer. In my eyes, all the abbreviations used here for the chemical compounds are very strange and difficult for remembering.

Answer: Following the suggest changes the principal component analysis has been cited without abbreviation in the manuscript. In addition, the first letter of line 73 has been changed.

COMMENT #3.Line 139-140, why the daughter ion of 299.9 was used for rutin? From my experience, the most abundant daughter ion for rutin would be 301. For the quercetin-3-O-glucoside, it is 301 too. For kaempferol 3-glucoside, it is around 285, like other glycosides of kaempferol

Answer: In our experimental set-up, when performing the optimization procedure, the signal from the ion at m/z 299.9 have more than double intensity with respect to the ion at m/z 301, using the own optimal collision energy for both ions. Generally, for all the components, we selected the most intense transition at its optimal collision energy value and a single transition since the qualitative confirmation takes advantage of the enhanced product ion full scan made in accordance with IDA experiments.

Reviewer 2 Report

Reviewer's comment on Manuscript Number: antioxidants-1586280

The manuscript entitled “Cocoa bar antioxidant profile enrichment with underutilized apple varieties” falls within the scope of Antioxidants.

The manuscript is certainly very interesting and valuable. This study could stimulate more research to further the exploration of the problem.  However, I have a few remarks:

  • The abstract needs to have more data about findings.
  • Please use more references to discuss statistical results. What is responsible for the diversification of samples?
  • What is the reason for the great variability in phenolic compounds?
  • Please check the English quality of the paper. It needs proofreading.

Author Response

COMMENT #1. The abstract needs to have more data about findings.

Answer: As suggested by Reviewer #2 the abstract has been improved with more data.

COMMENT #2. Please use more references to discuss statistical results. What is responsible for the diversification of samples? What is the reason for the great variability in phenolic compounds?

Answer: Apple varieties showed a great variability among the fifteen phenolic compounds studied. This aspect is well established in the literature and one possible explanation is the resistance to diseases or adverse environmental conditions among varieties (Kalinowska et al., 2014). Moreover, underutilised apples varieties are usually characterise by bitterness that is due to polyphenolic molecules (Ceci et al., 2021). In commercial variety, bitterness has been reduced favouring traits such as sweetness.

Kalinowska, M.; Bielawska, A.; Lewandowska-Siwkiewicz, H.; Priebe, W.; Lewandowski, W. Apples: Content of phenolic compounds vs. variety, part of apple and cultivation model, extraction of phenolic compounds, biological properties. Plant Physiology and Biochemistry 2014, 84:169-188

https://doi.org/10.1016/j.plaphy.2014.09.006.

Ceci, A.T.; Bassi, M.; Guerra, W.; Oberhuber, M.; Robatscher, P.; Mattivi, F.; Franceschi, P. Metabolomic Characterization of Commercial, Old, and Red-Fleshed Apple Varieties. Metabolites 2021, 11, 378. https://doi.org/10.3390/ metabo11060378

This aspect has been introduced in the manuscript (changes are in red).

COMMENT #4. Please check the English quality of the paper. It needs proofreading.

Answer: The English quality has been checked.

Round 2

Reviewer 1 Report

As I mentioned in the first-round review, “the discussion section is too weak, I cannot find any answer to my question which raised when I read the result part. For example, the content of quercetin in cultivar ‘Golden’ is much higher than other cultivars, why the cocoa bar added to this cultivar is not the highest one? In the same way, for DPPH scavenging activity, the ‘Panaia’ is the best one, but with cocoa mass, its activity became the weakest one? There are many questions between the compounds and activities. I think the authors need to think about those questions and try to give a reasonable explanation. “ In this version, I still cannot find the answer.
